# Harmful dimensions of medical culture in relation to physician burnout: A cross-sectional study

**Emilie Banse**[1,2]*, **Moïra Mikolajczak**[1], **Marie Bayot**[3‡], **Anne-Laure Lenoir**[3‡], **Philippe de Timary**[2]

**1** Department of Psychology, Psychological Sciences Research Institute, Catholic University of Louvain, Louvain-La-Neuve, Belgium, **2** Department of Adult Psychiatry, Cliniques Universitaires Saint-Luc, Institute of Neuroscience, Catholic University of Louvain, Brussels, Belgium, **3** Department of Clinical Sciences, University of Liège, University Hospital of the Sart Tilman, Liège, Belgium

☯These authors contributed equally to this work.
‡ MB and A-LL contributed equally to this work.
*emilie.banse@uclouvain.be

## Abstract

Physician burnout (PB) is an emotional exhaustion in response to long-term workplace stress. Despite growing interest in how the medical culture contributes to physician burnout, there is little empirical research on the subject. In this study, we aimed to evaluate quantitatively specific dimensions of the medical culture, thereby testing the hypothesis that a group of interdependent aspects of medical culture would positively correlate with PB. We collected on-line survey results in a cross-sectional study of 1002 physicians in current clinical practice. Participants completed the Burnout Assessment Tool and additional survey items measuring aspects of contemporary medical culture. Using exploratory and confirmatory factor analyses, we first explored and confirmed the factorial validity of a model representing the medical cultural dimensions, and then undertook regression analyses to investigate their associations with physician burnout. The analyses revealed seven relevant factors of the medical culture clustering in three distinct but interdependent broader dimensions, which we designated as (1) Physician's Professional Commitment, (2) The Myth of the Invulnerable Physician, and (3) Physician Stigma towards Burnout. These three dimensions had independent associations with increased burnout, suggesting that they are detrimental to the welfare of physicians. Additionally, we identified a specific factor reflecting the Existential Significance of Being a Physician, which mitigated against burnout. Upon controlling for sociodemographic and professional characteristics, the investigated dimensions of medical culture accounted for 30% of the variance in PB. This quantitative exploration of the relations between medical culture and physician defined dimensions of medical cultural that are harmful to physicians and by extension to patient care. Present results highlight the need for further empirical investigations of medical culture and the pathways whereby certain dimensions of medical culture specifically relate to the well-being and health of physicians.

**Data availability statement:** The data underlying the findings of this study are publicly available in the Open Science Framework (OSF) repository at DOI 10.17605/OSF.IO/VZ7QY (https://osf.io/vz7qy/).

**Funding:** EB is a research fellow of the Fonds de la Recherche Scientifique – FNRS (Belgium) and received funding under grant number 40010334. The funding agency had no role in the design, execution, interpretation, or publication of this study.

**Competing interests:** The authors have declared that no competing interests exist.

## 1. Introduction

Physician burnout (PB) is highly prevalent and of rising incidence across all medical specialties [1–4], bringing significant consequences for the health and welfare of physicians, and to the detriments of patient care and the effectiveness of healthcare institutions [5,6].

Scholars and physicians are increasingly advocating for a broader recognition that aspects of contemporary medical culture are important contributors to PB [7,8]. In general terms, a culture represents a set of shared beliefs within a social group, which shapes expectations for behaviors and fosters a sense of identity among group members [8,9]. The medical culture is a distinct and deeply ingrained professional culture [10], which consists of formal and informal elements acquired through socialization [11,12]. Aspiring physicians internalize prevailing attitudes, norms, and values during their training in the practice of medicine, which subsequently become implicit (and at times rigid) cultural tenets within the medical community [12].

The medical culture is commendable in many respects, being characterized by service, dedication, compassion, and an unwavering commitment to excellence and professional competence [8]. Physicians are profoundly motivated to attend to the needs and well-being of their patients, and the prevailing medical culture combines a strong dedication to healing with a continuous pursuit of progress in clinical care. These commonly held values maintain the practice of medicine as a highly respected and sought-after profession, and provide physicians with a profound sense of meaning and professional fulfillment [11,13]. Nonetheless, other specific aspects of medical culture may have detrimental consequences for the well-being and health of practicing physicians [8,11]. Through editorials and personal viewpoints [13–17], reviews [18,19], testimonials [20,21], and books [11,22,23], physicians are increasingly expressing concerns that aspects of medical culture negatively impact their well-being, thus contributing to what might be called a professional malaise. Such personal attestations aside, empirical research on medical culture and physicians' distress remains a nascent field of study. Few investigations have quantitatively investigated associations between PB and recognized deleterious aspects of the medical culture such as low self-compassion [24,25], the imposter phenomenon in the medical profession [26], as well as stigma and attitudes towards help-seeking by physicians in distress [27–30]. However, those studies placed their primary focus on isolated elements of medical culture, without considering the interconnection and synergism of combined elements. This approach limits our understanding of the role of beliefs, norms, and attitudes typical of the medical culture that could bear a relation to the phenomenon of PB. Moreover, the preponderance of such empirical studies were conducted in the United States [24–28], with no comparable studies yet conducted in Europe. Overall, research in this field remains at an inchoate stage, as compared to the better developed state of investigations of risk factors for PB encountered at individual, work-related, and organizational levels [31–37]. Consequently, prevention and intervention strategies for PB currently fail to address the detrimental contributions of medical culture [38].

On the other hand, there is an abundant qualitative literature highlighting various professional norms of medical culture that have not yet been captured by quantitative studies. For example, some studies have suggested that an expression by physicians of mental health concerns is tacitly equated with a lack of professional competence [7,23,39]. This misperception presents a unique challenge that shapes physicians' responses to their own distress (i.e., by delaying help-seeking) and that of their peers (i.e., the risk of ostracism) [40]. Other examples of medical culture include the expectation of invulnerability [41,42], personal commitment to the professional role and identity [43], or prioritization of work above personal well-being [39,44]. These deeply embedded norms likely reinforce each other. Although well-characterized in isolation, these norms have not yet been thoroughly quantified in relation to their roles in PB.

In response to this scenario, we designed the present study to evaluate quantitatively multiple dimensions of medical culture and their associations with PB. Based on an extensive literature review including the available qualitative studies, we selected a set of norms that are repeatedly cited by physicians and scholars as potential contributors to PB, which are presented in detail in Table A of S1 Tables. In our first aim, we explored and validated a model of specific professional norms within the medical culture that may relate to PB incidence. While this psychometric work represents an important methodological contribution of the study, our main goal was to examine the relationship between harmful aspects of medical culture and burnout. We hypothesized that the relevant dimensions would be interdependent and would show independent associations with PB.

## 2. Material and methods

### 2.1. Ethics statement

This cross-sectional study was approved by the Institutional Review Board (Ethics Committee of the Cliniques Universitaires Saint-Luc, Brussels) prior to data collection (Protocol Number: 2023/07MAI/219) and is a part of a larger research project preregistered on OSF (Preregistration). This report complies with the Checklist for Reporting of Survey Studies [45] and follows the Strengthening the Reporting of Observational Studies in Epidemiology (STROBE) reporting guideline for cross-sectional studies.

### 2.2. Survey instrument

An online survey was administered to participants, with anonymization to ensure participant confidentiality. Data collection and management was facilitated through REDCap electronic data capture tools [46]. Recruitment of participants occurred iteratively during the data collection phase, which lasted from September 5 to November 5, 2023. Participants provided written consent before completing the survey. Participants provided the following personal and professional characteristics: age, gender, relationship status, parental status, specialty, work-setting, years in practice, average weekly work hours, and country of residence. The professional norms of the medical culture were assessed using a questionnaire of 35 items developed by the research team (detailed in Table A of S1 Tables), which had previously been reviewed and refined by researchers and professionals specializing in the care of physicians suffering from burnout. Some items were adapted from the Stigma of Occupational Stress Scale for Doctors (SOSS-D) [29]. Responses were rated on a 5-point Likert scale ranging from "strongly disagree" (1) to "strongly agree" (5). Burnout was measured using the 12-item version of the Burnout Assessment Tool (BAT-12), a recently validated self-assessment questionnaire for measuring burnout [47,48]. We used the French version of the BAT-12, as proposed by the original authors of the scale [49]. The BAT-12 was developed to address certain conceptual, technical, and practical imperfections of the Maslach Burnout Inventory. It measures symptoms of exhaustion, mental distance, cognitive impairment, and emotional impairment. Responses were rated on a 5-point Likert scale ranging from "never" (1) to "always" (5). A global burnout score, ranging between 1 and 5, was calculated by averaging the item scores [47,48]. In the current sample, the Cronbach's' alpha (α) of the burnout scale was 0.88, indicating good interclass agreement.

### 2.3. Study population

The participants were French-speaking physicians in current clinical practice, encompassing general practitioners (GPs), specialist physicians (all specialties), and medical residents (postgraduates in training to become GPs or specialists). To construct a convenient sample

representative of the broad physician population, we used several approaches to encourage participation. Leading medical organizations in Belgium, general medicine organizations and physicians' magazines also relayed the survey, with some additional recruitment in France. Deans of medical faculties at French-speaking universities in Belgium shared the survey via their mailing lists for residents and academic physicians at their respective university hospitals. Medical directorates and councils of different hospitals shared the survey among physicians. The survey was also disseminated on social media networks and via word of mouth. While the initial recruitment strategy was focused on physicians practicing in Belgium, this online dissemination allowed for the participation of French-speaking physicians from other countries as well. Participants were not excluded based on their country of residence, as long as they met the inclusion criteria. Participation was voluntary and without financial incentive.

This study employed convenience sampling, and as such, no census was conducted. Consequently, we did not estimate the exact participation rate per country, nor did we track the number of potential participants by country. The study's design did not aim to perform country-specific analyses, as this was not within the scope of the research objectives. Nevertheless, we acknowledge that the inclusion of participants from different countries may have introduced variability related to differing healthcare systems, cultural norms, and medical practices, an issue that we discuss further in the discussion.

## 2.4. Statistical analysis

Physician burnout was defined as the outcome (dependent variable). The independent variable, or predictor, was the structure of medical culture. The main analyses are described below, with provision of details in S1 Text. Statistical analyses adhered to the preregistration, with the addition of a confirmatory factor analysis (CFA) to cross-validate the results of the exploratory factor analysis (EFA). According to our sample size calculation (described in S1 Text), a minimum of 550 participants was necessary for the planned analyses. However, the final sample size significantly surpassed this threshold, enabling the execution of both EFA and CFA. Descriptive statistics were computed for all variables, and patterns of drop-out and missing values were examined. Missing values were primarily due to participant dropout during the survey. Participants with complete data on the cultural items (N = 1002) were compared to those with missing data (N = 236), revealing only small differences, notably a higher likelihood of survey dropout among residents ($\chi 2(4) = 5.80$, $p = 0.02$, $\varphi = 0.070$). Potential differences were also examined between individuals who provided complete data on all items of interest (35 cultural items and 12 burnout items, N = 973) and those who provided complete data only on the cultural items (N = 1002) on demographic and work-related variables (age, gender, seniority, number of hours worked per week, status of the physician, specialty) as well as on burnout, using t-tests and $\chi 2$ tests. No significant differences were found between these groups, indicating that they were comparable with respect to these variables. To maximize statistical power, analyses were conducted on the largest possible samples. No imputation was performed for incomplete answers and drop-out and missing data were deleted listwise.

To first ascertain and then cross-validate the factor structure of the investigated dimensions of the medical culture, the sample was randomly divided into two subsamples of 501 participants assigned for either EFA or CFA. Using $\chi 2$ analyses for categorical variables, and t-tests and one-way ANOVAs for continuous variables, we first verified the comparability of the subsamples regarding sociodemographic characteristics and burnout scores. In the first subsample, we conducted EFA using principal axis factoring (PAF) with direct oblimin rotation on the 35 items [50–53]. We assessed suitability of our data with the Kaiser-Meyer-Olkin (KMO) measure of sampling adequacy and Bartlett's test of sphericity. Item loadings of ≥ 0.40 and communalities > 0.20 were required; cross-loading items were defined as those items loading

≥ 0.30 on two or more factors. Such items were considered unstable and were therefore removed. The proportion of total variance explained (TVE) by the factors needed to be ≥ 50%. To determine the number of factors to retain, we relied on various methods: Kaiser's criterion, the scree plot, and the parallel analysis [50–53]. We performed sensitivity analyses to test the stability of factor solutions (e.g., after removing outliers or changing factor retention criteria). These analyses supported the robustness of the identified factor structure.

Subsequently, in the second subsample, we conducted a CFA with the maximum likelihood estimation method to validate the structure emerging from the EFA. We compared five statistical models (outlined in Fig A through Fig E of S1 Fig and described in Table E of S1 Tables), using different goodness-of-fit indices to determine the statistical acceptability of the models and to retain the best-fitting model: the chi-square indices ($\chi2$) [54], the root mean square error of approximation (RMSEA), the standardized root mean square residual (SRMR), the comparative fit index (CFI), and the Tucker-Lewis index (TLI) [55]. Acceptable fit criteria included RMSEA and SRMR ≤ 0.08, CFI and TLI > 0.90 [54,55]. After comparing various models, we retained the best-fitting model [54,55], yielding what we designate as harmful dimensions of medical culture (henceforth HDMC).

We conducted further analyses with this model on the entire sample, including descriptive statistics and assessment of internal consistency (Cronbach's α coefficients). We computed one-tailed correlation tests (Spearman correlations to account for non-normal data distribution) between the HDMC and PB scores, with significance level set at $p < 0.001$ after adjustment for multiple testing. A hierarchical multiple regression (two-block regression model) was then conducted to assess the additional variance in burnout scores explained by medical culture. The hierarchical regression analysis included the following potential confounders in the first step of the model: gender, relationship status, parental status, physician status (resident vs attending), specialty (generalist vs specialist), years in practice, and weekly working hours, as these personal and professional characteristics are known to be associated with PB [31,35,37]. In step two, we included the investigated dimensions of medical culture. Statistical analyses were performed using IBM SPSS 28.0 [56], except CFAs, which were conducted using Stata 18 Software [57].

## 3. Results

The main findings are summarized below. Comprehensive results are available in Table B to K of S1 Tables.

### 3.1. Participants

The final sample comprised those participants (N = 1002; 64.6% women) who had provided complete data for all survey items. The sociodemographic and work-related characteristics of the 1002 surveyed physicians, including detailed subsets for specialist physicians and general practitioners, are detailed in Table 1. Data are reported as number (percentage). The median age was 38 years (IQR, 30-50). 82.3% reported being in a relationship and 41.4% indicated having children. Nearly all respondents (92%) resided in Belgium, with the remainder in France and Switzerland. 27.7% of the sample were GPs, 6.1% were resident GPs, 45.6% were specialists, and 18.8% were resident specialists. A small proportion of respondents (1.5%) were either occupational, insurance, school physicians, or medical consultants. Among the 645 specialists, the most common specialties were anesthesiology, pediatrics, and surgery. 86.6% of the specialists and resident specialists worked in a hospital setting, largely at university hospitals (46.5%). On the other hand, 44.2% of the participating GPs worked in multidisciplinary groups. The median duration of practice was 12 years (IQR, 5-24) and the median estimated weekly working hours was 50 (IQR, 40-60). The median burnout score was 2.08 (IQR, 1.75-2.42).

**Table 1. Sociodemographic and work-related characteristics of the 1002 surveyed physicians, including detailed subsets for specialist physicians and general practitioners.**

| Sociodemographic and work-related characteristics (total sample, N = 1002) | N (%) |
|---|---|
| **Gender** | |
| Male | 347 (34.6) |
| Female | 547 (64.6) |
| Other | 2 (0.2) |
| Missing | 6 (0.6) |
| **Relationship status** | |
| Single | 175 (17.5) |
| In a relationship | 825 (82.3) |
| Missing | 2 (0.2) |
| **Parental status** | |
| Yes | 582 (41.4) |
| No | 415 (58.1) |
| Missing | 5 (0.5) |
| **Country of residence** | |
| Belgium | 922 (92) |
| France | 65 (6.5) |
| Switzerland | 2 (0.2) |
| Other | 6 (0.6) |
| Missing | 7 (0.7) |
| **Type of physician** | |
| General practitioner | 339 (33.8) |
| Specialist physician | 646 (64.5) |
| Occupational medicine physicians/ Insurance physicians/ School physicians/ Medical consultants | 14 (1.4) |
| Missing | 3 (0.3) |
| **Work-related characteristics (specialist physicians only, N = 646)** | N (%) |
| **Status** | |
| Attending | 458 (70.9) |
| Resident | 188 (29.1) |
| **Work setting** | |
| Hospital setting | 560 (86.6) |
| Private consultation | 13 (2.0) |
| Both | 60 (9.3) |
| Missing | 13 (2.0) |
| **Type of hospital** | |
| Public hospital | 204 (32.9) |
| Private hospital | 127 (20.5) |
| University hospital | 288 (46.5) |
| Missing | 27 (4.2) |
| **Specialty** | |
| Anesthesiology | 60 (9.3) |
| Pediatrics | 52 (8.1) |
| Surgery | 49 (7.6) |
| Acute and emergency medicine | 45 (7.0) |
| Neurology and pediatric neurology | 40 (6.2) |
| Psychiatry | 37 (5.7) |

*(Continued)*

**Table 1.** (Continued)

| Sociodemographic and work-related characteristics (total sample, N = 1002) | N (%) |
|---|---|
| Internal medicine | 35 (5.4) |
| Gynecology | 25 (3.9) |
| Intensive care | 24 (3.7) |
| Cardiology | 19 (2.9) |
| Gastro-enterology | 19 (2.9) |
| Geriatrics | 18 (2.8) |
| Otolaryngology | 18 (2.8) |
| Pneumology | 18 (2.8) |
| Hematology | 16 (2.5) |
| Physical medicine and rehabilitation | 15 (2.3) |
| Dermatology | 13 (2.0) |
| Endocrinology | 13 (2.0) |
| Radiology | 13 (2.0) |
| Clinical biology | 11 (1.7) |
| Clinical infectiology | 11 (1.7) |
| Oncology | 11 (1.7) |
| Ophthalmology | 10 (1.6) |
| Urology | 9 (1.4) |
| Nuclear medicine | 8 (1.2) |
| Nephrology | 8 (1.2) |
| Pathological anatomy | 7 (1.1) |
| Neurosurgery | 6 (0.9) |
| Radiotherapy – oncology | 5 (0.8) |
| Rheumatology | 3 (0.5) |
| Neonatology | 2 (0.3) |
| Stomatology | 2 (0.3) |
| Palliative care | 2 (0.3) |
| Other (hospital hygiene, medical information, orthodontics, legal medicine) | 4 (0.6) |
| Missing | 18 (2.8) |
| **Work-related characteristics (general practitioners only, N = 339)** | **N (%)** |
| **Status** | |
| Attending | 278 (82.0) |
| Resident | 61 (18.0) |
| **Type of practice** [a] | |
| General medicine (consultations and home visits) | 327 (96.5) |
| Hospital setting | 14 (4.1) |
| Other institution (e.g. family planning center, nursing home) | 52 (15.3) |
| Non-conventional medicine | 11 (3.2) |
| **Work setting** [a] | |
| Works alone | 90 (26.5) |
| Monodisciplinary group | 107 (31.6) |
| Multidisciplinary group | 150 (44.2) |

N of the total sample = 1002, N of the subset of specialist physicians = 646, N of the subset of general practitioners = 339. [a] Multiple responses were possible.

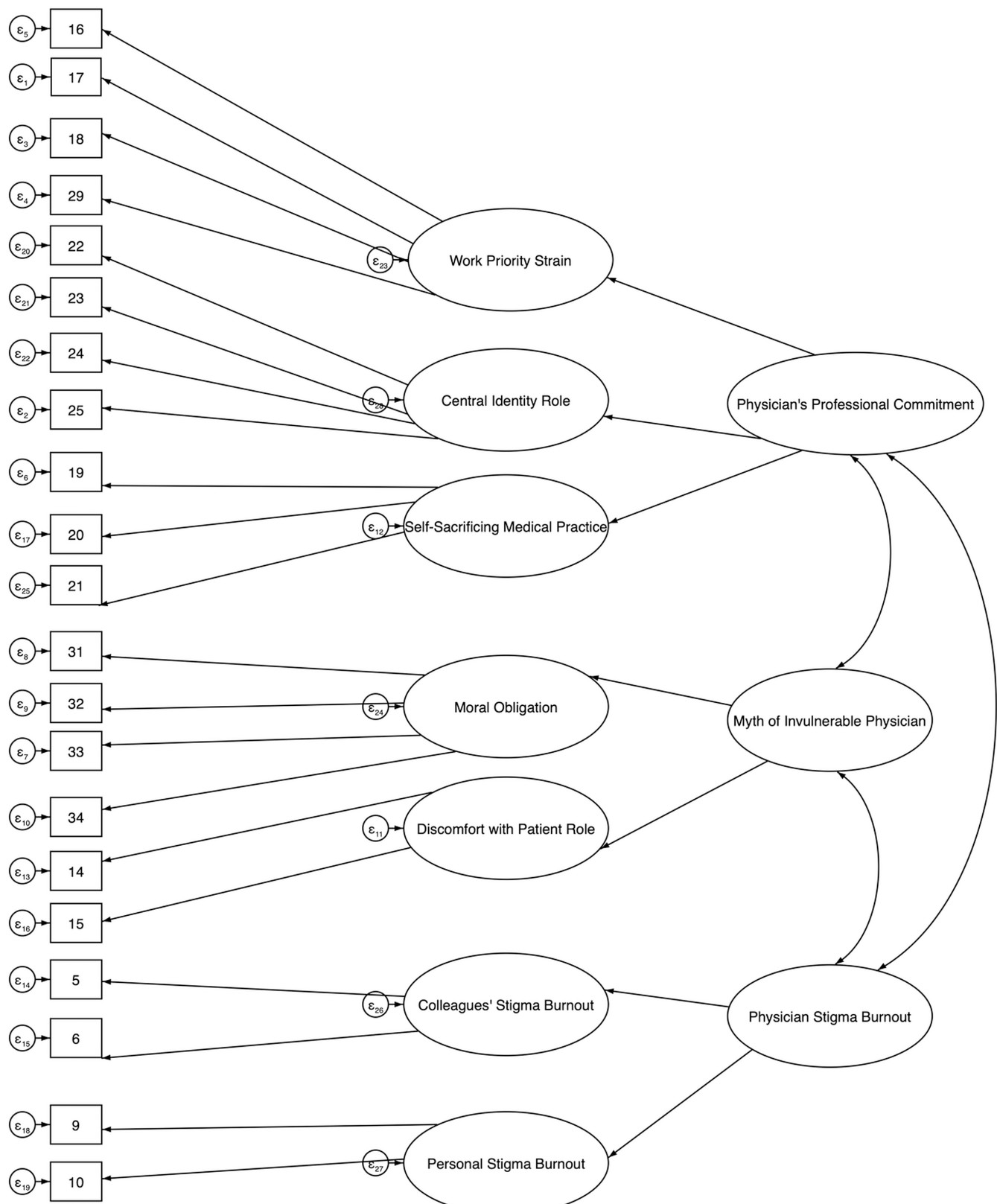

**Fig 1. Visual representation of the final Harmful Dimensions of the Medical Culture Model.** In this figure, the top left column represents the error (e) terms fit in the CFA. The second column (rectangles) starting from the left represents the items (21 in total). The third column (arrows) represents the

item loadings onto the first order latent factors. The fourth column (ovals) represents the first order latent factors and their associated error (e) terms. The fifth column (arrows) represents the structural loadings from the first order latent factors to the second order latent factors. The last column (ovals) represents the second order latent factors with inter-factor correlations (arrows). This is a second-order hierarchical model in which seven of the eight first-order factors of the EFA load onto three second-order dimensions (HDMC).

**Table 2. Description of the Harmful Dimensions of the Medical Culture.**

| Second-order dimensions with definition | First-order factors | # of items | α | Median (IQR) |
|---|---|---|---|---|
| **Physician's Professional Commitment** | | 11 | 0.85 | 3.72 [3.27;4.09] |
| Higher scores on this dimension reflect the physician's tendency to consider their medical profession as central to their manner of defining their identity and daily lives. This reflects the sacrifices made for the medical career and work-life imbalance often arising in the medical profession. | Work Priority Strain | 4 | 0.87 | 4.00 [3.50;4.50] |
| | Physician's Central Identity Role | 4 | 0.78 | 3.00 [3.50;4.50] |
| | Self-Sacrificing Nature of Medical Practice | 3 | 0.86 | 4.33 [4.00;5.00] |
| **The Myth of the Invulnerable Physician** | | 6 | 0.77 | 3.83 [3.16;4.16] |
| Higher scores reflect the physician's tendency to perceive that physicians should be invulnerable, and the roles of physicians and patients as being incompatible. This reflects an inclination towards believing that physicians should not themselves experience illness. | Physician's Moral Obligation to Patients and Colleagues | 4 | 0.78 | 3.75 [3.00;4.25] |
| | Physician's Discomfort with Patient Role | 2 | 0.72 | 4.00 [3.00;4.50] |
| **Physician Stigma towards Burnout** | | 4 | 0.62 | 3.45 [3.05;4.80] |
| Higher scores on this dimension reflect the physician's proneness to negatively perceive the experience of burnout in physicians. This tendency is reflected by the physician's beliefs about stigmatizing attitudes held by his or her colleagues, and by their internalized stigmatizing attitudes towards burnout in the medical profession. | Colleagues' Stigma towards PB | 2ᵃ | 0.84 | 3.00 [2.00;4.00] |
| | Personal Stigma towards PB | 2 | 0.68 | 2.00 [1.50;3.00] |
| **HDMC global score** | | 21 | 0.85 | 3.45 [3.05;4.80] |
| Higher global score reflects a general tendency of the physician to endorse the harmful dimensions of the medical culture. | | | | |

N = 1002. α = Cronbach's alpha, to assess the internal consistency of the scale. IQR = Interquartile Range. This table only includes the HDMC factors. ᵃ For testing internal consistency, we conducted inter-item correlation when only 2 items constituted a scale.

## 3.2. Exploratory and confirmatory factor analysis

Depending on the criteria, there were seven to ten extractable factors from the initial EFA. We ran multiple EFAs, iteratively removing items until achieving a stable solution meeting all preestablished statistical criteria. The final structure comprised eight factors including 24 items. The Kaiser-Meyer-Olkin (KMO) measure of sampling adequacy of this structure was high (0.84), and Bartlett's test of sphericity was highly significant, $\chi^2$ (276) = 5069.83, p < 0.000. This solution accounted for 59% of item variance and yielded a simple structure (with no cross-loading items). Based on item content, we designated the factors as follows: (1) Work Priority Strain, (2) Self-sacrificing Nature of Medical Practice, (3) Physician's Central Identity Role, (4) Physician's Discomfort with Patient Role, (5) Physician's Moral Obligation to Patients and Colleagues, (6) Colleagues' Stigma towards PB, (7) Personal Stigma towards PB, and (8) Existential Significance of Being a Physician.

CFA allowed the comparison of the EFA structure with alternative models using various fit indices (see Fig A to E in S1 Fig and Table E in S1 Tables for depictions of these models and their

**Table 3. Correlations between factors of the Harmful Dimensions of the Medical Culture (HDMC) and PB.**

| | First-order factors | | | | | | | Second-order dimensions | | | Burnout |
|---|---|---|---|---|---|---|---|---|---|---|---|
| | 1 | 2 | 3 | 4 | 5 | 6 | 7 | A | B | C | |
| 1. Work Priority Strain | -- | | | | | | | **0.82**\*\* | 0.38\*\* | 0.20\*\* | *0.40*\*\* |
| 2. Physician's Moral Obligation to Patients and Colleagues | 0.37\*\* | -- | | | | | | 0.40\*\* | **0.90**\*\* | 0.26\*\* | *0.33*\*\* |
| 3. Colleagues' Stigma towards PB | 0.19\*\* | 0.27\*\* | -- | | | | | 0.21\*\* | 0.28\*\* | **0.82**\*\* | 0.23\*\* |
| 4. Personal Stigma towards PB | 0.08 | 0.09\* | 0.13\*\* | -- | | | | 0.14\*\* | 0.18\*\* | **0.65**\*\* | 0.06 |
| 5. Physician's Discomfort with Patient Role | 0.21\*\* | 0.34\*\* | 0.19\*\* | 0.14\*\* | -- | | | 0.25\*\* | **0.68**\*\* | 0.24\*\* | 0.23\*\* |
| 6. Physician's Central Identity Role | 0.35\*\* | 0.26\*\* | 0.11\*\* | 0.13\*\* | 0.16\*\* | -- | | **0.75**\*\* | 0.26\*\* | 0.16\*\* | 0.16\*\* |
| 7. Sacrificial Nature of Medical Practice | 0.57\*\* | 0.31\*\* | 0.25\*\* | 0.11\*\* | 0.22\*\* | 0.31\*\* | -- | **0.72**\*\* | 0.33\*\* | 0.26\*\* | *0.36*\*\* |
| A. Physician's Professional Commitment | | | | | | | | -- | | | *0.38*\*\* |
| B. The Myth of the Invulnerable Physician | | | | | | | | 0.42\*\* | -- | | *0.35*\*\* |
| C. Physician Stigma towards Burnout | | | | | | | | 0.25\*\* | 0.30\*\* | -- | 0.22\*\* |
| **HDMC global score** | | | | | | | | 0.84\*\* | 0.75\*\* | 0.56\*\* | *0.42*\*\* |

Correlations are Spearman correlations (*ρ*). \*Correlation is significant at p<0.01 (1-tailed). \*\*Correlation is significant at p<0.001 level (1-tailed), following adjustment for multiple testing. Inter-factor correlations are conducted among the entire participant group (N = 1002), and correlations with burnout were conducted among N = 996 participants. Inter-factor correlations are indicated in bold when a first-order factor is a part of the second order factor of the column. Correlations with burnout indicated italics are considered as being clinically meaningful, as they are of moderate effect size (Cohen's d ≥ 0.5, Spearman correlation coefficient ≥ 0.24).

corresponding fit indices). All models demonstrated satisfactory fittings to the data, but the hierarchical model displayed in Fig 1 emerged as the best model, in theoretical and statistical terms (RMSEA = 0.06 90% CI = [0.05 to 0.06]; CFI = 0.92; TLI = 0.91; SRMR = 0.05). In this model, seven of the eight first-order factors formed three second-order dimensions, with omission of factor eight, Existential Significance of Being a Physician from the model, as its inclusion reduced the overall fit. That initial finding suggests that factor eight may not align with the structure of *Harmful* Dimensions of the Medical Culture (HDMC), referring to our hypothesis to be validated by examining the correlations between the dimensions and burnout, as discussed below. The second-order dimensions, labeled (A) Physician's Professional Commitment, (B) The Myth of the Invulnerable Physician and (C) Physician Stigma towards Burnout, were informed by intercorrelations among first-order factors. This hierarchical model provides a concise representation of a complex structure, delineating distinct yet interconnected HDMC components.

### 3.3. Description and reliability of the HDMC

Table 2 describes the final HDMC along with descriptive statistics and internal consistency. As evidenced by the median scores and interquartile ranges (IQR), physicians exhibit strong endorsement of the HDMC (for items, see Table K in S1 Tables).

### 3.4. Interdependency of the HDMC and correlations with burnout

As shown in Table 3, the first- and second-order HDMC had varying degrees of positive correlation. Table 3 also illustrates the correlations between the first-order factors and second-order dimensions, thereby affirming the relevance of all three dimensions. Strong positive correlations were found between the global HDMC score and the dimensions of Professional Commitment (*ρ* = 0.84, p< 0.001), and the Myth of the Invulnerable Physician (*ρ* = 0.75, p< 0.001), respectively, suggesting their central role in the concept of harmful medical culture. In contrast, Physician Stigma towards Burnout showed a significant but notably lower correlation (*ρ* = 0.56, p< 0.001) suggesting that, while still important, it represents a less central dimension compared to the others two.

**Table 4. Hierarchical multiple regression of the association between the investigated dimensions of the medical culture and burnout, while controlling for covariates.**

| | Std. B | 95% CI for Std.B | p |
|---|---|---|---|
| $F_{(11, 963)} = 36.584$, p <.001, $R^2 = 0.297$, $R^2_{adjusted} = 0.289$ | | | |
| (Constant) | | | <.001 |
| Gender | .05 | 0.00 to 0.11 | 0.06 |
| Relationship status | -0.01 | -0.07 to 0.05 | 0.71 |
| Parental status | -0.02 | -0.10 to 0.06 | 0.65 |
| Physician status | 0.04 | -0.04 to 0.11 | 0.34 |
| Specialty | -0.06 | -0.12 to 0.00 | 0.04 |
| Years in practice | 0.05 | -0.02 to 0.14 | 0.24 |
| Hours worked per week | -0.03 | -0.11 to 0.05 | 0.39 |
| Physician's Professional Commitment | 0.38 | 0.31 to 0.44 | <0.001 |
| The Myth of the Invulnerable Physician | 0.22 | 0.15 to 0.28 | <0.001 |
| Physician Stigma towards Burnout | 0.08 | 0.03 to 0.14 | 0.004 |
| Existential Significance of Being a Physician | -0.31 | -0.37 to -0.26 | <0.001 |

N = 964. The dependent variable is the Burnout score. Std B = standardized beta. CI = Confidence Interval. Missing values excluded listwise. In step 1 of the model, we included the following covariates as referents: female gender, being in a relationship, having children, being a resident, being a specialist, years in practice and working hours estimated per week. In step 2, we added the HDMC and Existential Significance. Excluding the factor Existential Significance of Being a Physician, the three second-order HDMC significantly contributed to the prediction of burnout, while controlling for demographic and professional characteristics ($R^2 = 0.21$).

Except for Personal Stigma towards PB, all first-order factors exhibited strong positive correlations with burnout scores (p<0.001) with varying effect sizes. These correlations suggest that the greater endorsement of each dimension by physicians corresponds to higher burnout scores. There were positive correlations of moderate effect size between burnout scores and the second-order dimensions, providing empirical support for our hypotheses.

### 3.5. Distinctiveness of existential significance of being a physician

Existential Significance of Being a Physician ($\alpha = 0.76$) exhibited a positive correlation with Physician's Central Identity Role ($\rho = 0.44$, p< 0.001), but minimal to negligible correlations with the other factors. Furthermore, Existential Significance of Being a Physician demonstrated relatively weaker correlations with the second-order dimensions ($\rho_{D1} = 0.28$, p< 0.001; $\rho_{D2} = 0.17$, p< 0.001; $\rho_{D3} = 0.04$, p= 0.09). Removing this factor from the HDMC structure resulted in better fitting of the data. Moreover, unlike the HDMC, Existential Significance was *negatively* correlated to burnout, albeit with a small effect size ($\rho = -0.14$, p< 0.001). This result supported our decision to exclude the factor from consideration among *Harmful* Dimensions of the Medical Culture.

### 3.6. Relative contribution of the HDMC and existential significance of being a physician to PB

The factors gender, relationship status, parental status, physician status, specialty, years in practice, and working hours together accounted for less than 4% of the variance in burnout scores (see Table I of S1 Tables for the full two-block regression model). The HDMC and

Existential Significance accounted for an incremental 26% of the variance in burnout ($R^2$ = 0.30, adjusted $R^2$ = 0.29, F(11, 963) = 36.58, p < 0.001), as shown in Table 4. The three HDMC were independently associated with PB. Physician's Professional Commitment exhibited the strongest effect (Std B = 0.38, p < 0.001). Existential Significance was negatively associated with PB (Std B = -0.31, p < 0.001). Table I of S1 Tables presents the regression model upon controlling for social desirability.

## 4. Discussion

### 4.1. Summary of key findings

In our quantitative exploration of professional norms in medicine, we identified eight factors related to professional burnout (PB) in a large sample of mainly Belgian physicians. Seven of the eight factors regrouped into three dimensions of medical culture: Physician's Professional Commitment, the Myth of the Invulnerable Physician, and Physician Stigma towards Burnout. These dimensions were interdependent and positively correlated with PB, leading to their designation as Harmful Dimensions of Medical Culture (HDMC). However, the factor Existential Significance of Being a Physician did not fit in the HDMC structure, but rather demonstrated an inverse correlation with PB. The associations between the investigated dimensions and PB scores survived multivariable analysis adjusting for gender, relationship status, parental status, physician status, specialty, years in practice, and working hours. Results of this study extend previous empirical work on aspects of the medical culture by identifying new (sub-) dimensions of medical culture and exploring their interrelations and associations with PB.

### 4.2. Investigated dimensions of medical culture

The first dimension of HDMC, Physician's Professional Commitment, underscores the significant investment of physicians in their professional role, and their profound dedication, which together bring a risk for work-life imbalance [23,58,59]. This dimension reflects how the practice of medicine is a vocational calling that frequently intertwines personal and professional identities [59]. Physicians often feel compelled to define themselves and their value in relation to their working persona [23,60], such that self-sacrifice for the sake of the profession often becomes a cultural imperative, even when still in medical school [61]. While this commitment to medicine is commendable, it can become emotionally burdensome for physicians. The second HDMC, the Myth of the Invulnerable Physician, encapsulates perceptions of the need for invincibility [62] and the incompatibility between the separate roles of physician and patient [39]. Within a professional culture that assumes invulnerability, even the concept of illness becomes unthinkable [7]. Unfortunately, this notion is perpetuated through the process of medical education and is exacerbated by the real world contingencies of medical practices and organizations [23]. The third HDMC, Physician Stigma towards Burnout, embodies negative attitudes, either in self-assessment or in the view of medical peers, towards physicians who are facing professional challenges, and the resultant perceptions of being flawed or outsiders to the prestigious medical community [19,20,40,63]. Among physicians, stigma may be characterized by attitudes and processes that are unique to the profession [29].

The interdependency between the dimensions of Professional Commitment, Myth of the Invulnerable Physician, and Stigma towards Burnout suggests that the HDMC represents the composite of interconnected aspects of a broader medical culture; it is thus likely that the HDMC components reinforce one another. We found strong positive correlations between the global HDMC score and the dimensions of Professional Commitment and the Myth of the Invulnerable Physician, underscoring their centrality to the concept of harmful medical

culture. In contrast, Physician Stigma towards Burnout showed a weaker correlation. This may partly stem from methodological factors, such as the smaller number of items and its lower internal consistency, which may have contributed to greater variability and weaker correlations. Theoretically, stigma towards burnout may be less central to harmful medical culture than the broader constructs of professional commitment and invulnerability, which are historically deeply ingrained in the profession. However, our respondents' overall endorsement of the unofficial professional norms, such as invulnerability and professional commitment, highlights their pervasive nature in medical culture. Such norms are not unique to the field of medicine, but can also occur among members *inter alia* of the priesthood [23] or military [22]. Nevertheless, the demanding nature of medical training and work and the societal expectations placed upon physicians can amplify their impact [22]. Often internalized by physicians early in their medical training, the norms can become implicit over time, due to pervasive external reinforcement throughout a physician's career and professional environment.

While the investigated dimensions of medical culture may ill-serve the interests of physicians, and become detrimental to the care of patients [11], they are manifestly advantageous to the healthcare system, and often align with organizational culture in medicine [8]. Indeed, physicians experience increasing pressure to continuously meet growing patient demands [64] and workloads [65] without faltering. Institutions demand that physicians should function in systems based on external accountability [66] and adapt to new clerical burdens and digital health record systems [67]. Physicians who seek alleviation from these stressors by taking time off, working part-time, or admitting their vulnerability, often suffer penalties for these behaviors, which are perceived as manifestations of professional detachment [7]. As a result, these norms tend to become even more entrenched and difficult to address.

The factor Existential Significance of Being a Physician comprises items describing the extent to which a physician finds life-meaning through the practice of medicine. Physicians derive various aspects of meaning from their work (i.e., being a healer, developing a specific expertise, teaching) [68], which is influenced by their personal and professional values [69,70]. Since this factor is protective against burnout, we exclude it from the HDMC.

### 4.3. Associations with burnout

Our results show a strong association between PB and the individual endorsement of the HDMC, after controlling for characteristics associated with PB such as years in practice, gender, and physician status. This aligns with previous research on specific aspects of medical culture, including low self-compassion and the imposter phenomenon [24–26], which showed independent associations with burnout. Thus, aspects of the professional culture likely contribute to PB irrespective of personal and professional characteristics [25].

These results also support observations by physicians and researchers suggesting that medical culture may unintentionally contribute to burnout by perpetuating beliefs, values, and attitudes that are detrimental to physicians' health. Scholars have even argued that PB is an inevitable consequence of medical training and its encouragement of maladaptive behaviors [71,72]. The professional norms of medical culture investigated herein are also likely related to other prevalent health issues among physicians, including depression [1], suicidal ideation [73], and poor self-care behaviors [74] such as delayed help-seeking, which previously research has linked to stigmatization in physicians [27,28].

Deriving existential meaning from the engagement in the medical profession is consistently associated with lower burnout, even when controlling for other dimensions of medical culture. Previous research underscores that finding purpose and fulfillment in work can buffer against burnout in physicians [68] and enhance their psychological well-being [75]. However,

given the cross-sectional design of our study, we cannot infer causal relationships among our findings. Perceiving the practice of medicine as a source of meaning in life could shield physicians from burnout. Conversely, physicians experiencing burnout may cease to obtain life-meaning from engagement in their profession.

### 4.4. Limitations and future directions

By design, this investigation emphasizes the harmful aspects of medical culture, whereas other aspects of the professional culture of physicians may have positive effects. Many laudatory aspects of physicians' professional culture contribute to finding meaning and professional fulfillment, and reduced burnout risk. Physicians can exemplify qualities such as courage, dedication, altruism, and compassion [8,11,13], which are foundational to the medical profession and deeply ingrained in its culture. Future studies should explore the complex interplay between the positive and negative dimensions of medical culture and how they relate to physicians' well-being.

Due to the complex recruitment strategy, we are unable to estimate the net response rate for participation, which may raise caveats about the generalizability of our findings. For instance, the notably lower median age of our participants compared to the average age of the overall physician population in Belgium may suggest a self-selection bias. Also, the predominance of hospital-based specialty physicians may have influenced the findings. Future studies could benefit from a more diverse sample that includes a broader representation of age groups, work settings, and career stages within the physician workforce.

While the present study primarily included Belgian physicians, the inclusion criteria allowed participation from French-speaking physicians practicing in other countries. This inclusion introduced the possibility of variations in general culture, healthcare systems, and local medical cultures between countries, which could have influenced our findings. While this diversity could be viewed as a limitation, as we were unable to account for potential differences linked to these contextual factors, in our assessment it represents a strength, as it permits exploration beyond a single national context. The consistent findings support the hypothesis that a shared medical culture exists across countries, characterized by professional norms and expectations instilled during medical training and professional practice, and that this culture significantly contributes to PB irrespective of national variations.

The investigated aspects of medical culture mainly encompassed a triad of dimensions showing a positive correlation with PB. However, our study did not explicitly investigate how medical culture and its association with burnout may vary according to various personal (e.g., personality, generation) and socio-professional (e.g., status, specialty, work setting) characteristics. For example, physicians in academic hospitals may face additional demands related to teaching and research, which might amplify certain aspects of medical culture, such as professional commitment [3,43]. Similarly, the realities of specific specialties may interact with cultural norms differently, thereby influencing their impact on burnout. A more nuanced future study might explore how these characteristics influence the endorsement of the HDMC, and how such characteristics moderate the association between the HDMC and PB. We consider it likely that intricate interactions among physician characteristics and their endorsement of specific aspects of medical culture would moderate the outcome of PB.

The cross-sectional design of the study precludes us from making causal inferences. Further investigations with longitudinal designs seem warranted to establish causal relationships among HDMC and PB. We also see a need to examine the relative contributions of the HDMC and known PB risk factors, such as aspects of personality and organizational risk factors. Future studies might also expand on the psychometric work initiated in this study. It would be particularly valuable to extend the examination of the criterion and predictive validity of the scale, notably by investigating its impact on variables related to physicians' health and

well-being (e.g., depression, anxiety, reluctance to seek help, and self-medication), as well as team-level outcomes (e.g., quality of the work climate, interpersonal conflicts, workload distribution, and team collaboration) and patient-level factors (e.g., diminished empathy, medical errors, or, conversely, enhanced dedication to patient care and continuity of care). These prospects for future studies could provide a more comprehensive understanding of the impact of harmful dimensions of medical culture (HDMC), and illustrate how these dimensions may exert differential effects, including potentially positive outcomes in specific contexts.

Present findings have broad implications for medical education, burnout prevention in students and physicians, and tailored interventions for PB. Demonstration of a causal relationship whereby HDMC leads to PB would call for changing the culture of medicine to prevent or mitigate burnout in the profession. This scenario aligns with physicians' perspectives reported in the literature, which highlight the urgent need to direct investigations towards the unhealthy culture of medicine [19], to start a "cultural revolution"[20], and to rethink the professional culture of medicine [8]. While many stressors may be inherent to the medical profession, those embedded within the contemporary medical culture might be alterable [13] to the benefit of the medical community [19] and patients [11].

We perceive a need for nuanced strategies to address certain harmful cultural norms, while promoting the factors that enhance personal fulfillment in the medical profession. Such an initiative would by no means interfere with the knowledge, clinical skills, and ethical standards of physicians, but would promote a professional culture that is more attentive to the well-being of its members. Arguably, the time has come to reconcile the standards of excellence in the practice of medicine with a consideration of physicians' health. Unseating the deleterious professional norms identified in the present study calls for measures such as critically examining the detrimental aspects of the hidden curriculum in medicine [12], fostering role models and supervisors who do not conceal their vulnerability [76,77], reducing stigma through education and other pro-active initiatives [78], and initiating open discussions in the medical community on medical culture. Targeted interventions should also address organizational and systemic elements that perpetuate and derive benefit from the harmful aspects of the medical culture [79].

## 5. Conclusions

Physicians embody deeply entrenched positive values at the core of medical culture. However, the phenomenon of PB calls for an imperative to acknowledge the beliefs, norms, and attitudes prevailing in medical culture that adversely affect physicians' well-being. Present results underscore the significant relationships between three cultural dimensions—Physician's Professional Commitment, the Myth of the Invulnerable Physician, and Physician Stigma towards Burnout—and the risk of PB. These dimensions emphasize the need for targeted interventions to mitigate PB and promote physician well-being. Engaging in a meaningful reflection upon the need for cultural adaptation is a crucial step towards reconciling the dual nature of medical culture. By making a concerted effort towards embracing its positive attributes with respect to life-meaning, while addressing harmful norms of medical culture, physicians could safeguard the sustainability of their profession in the face of increased challenges in the workplace.

## Supporting information

**S1 Tables.   Table A. Identified potentially harmful professional norms of the medical culture following the review of literature with respective items and references.** Items 4 to 10 are items of the SOSS-D (The Stigma of Occupational Stress Scale for Doctors)[29]. **Table B. Main parameters of the final EFA solution.** This table includes loading parameter estimates,

Cronbachs' alpha (α), Eigenvalues and Variance explained. N = 501. 24 items. Factor loadings in EFA> |0.40| are presented. For EFA; F1 = Work Priority Strain, F2 = Existential Significance of Being a Physician, F3 = Physician's Moral Obligation to Patients and Colleagues, F4 = Colleagues' Stigma towards PB, F5 = Personal Stigma towards PB, F6 = Physician's Discomfort with Patient Role; F7 = Physician's Central Identity Role, F8 = Sacrificial Nature of Medical Practice. **Table C. Communalities of the final EFA solution before and after extraction.** Extraction Method: Principal Axis Factoring. **Table D. Spearman rho's correlations (ρ) between the EFA factors.** in subsample 1 (N = 501). *The correlation is significant at p<0.01 (1-tailed). **The correlation is significant at p<0.001 (1-tailed). One-tailed correlations were privileged, as we suspected positive correlations between the factors, representing distinct but interdependent dimensions of the same underlying concepts (harmful dimensions of the medical culture). **Table E. Description and model fit indices for the different HDMC models.** N = 501. χ2 = chi-squared; Df = degrees of freedom; CFI = comparative fit index; TLI = Tucker–Lewis index; RMSEA = root mean square error of approximation; SRMR = standardized root mean square residual. Model 1 is the model that was derived from the EFA. Model 5 is the final retained model. **Table F. Standardized structural coefficients (CFA) for the final HDMC Model.** Results are presented along with their respected 95% confidence intervals and reliability estimates, estimated in the total sample (N = 1002) and on 21 items. All standardized coefficients are significant at p-value <0.000. F1 = Work Priority Strain, F2 = Physician's Moral Obligation to Patients and Colleagues, F3 = Colleagues' Stigma towards PB, F4 = Personal Stigma towards PB, F5 = Physician's Discomfort with Patient Role, F6 = Physician's Central Identity Role, F7 = and Sacrificial Nature of Medical Practice. **Table G. Standardized covariances of the second-order dimensions for the final HDMC Model.** Results are presented along with their respected 95% confidence intervals (N = 1002). **Table H. Standardized measurements coefficients (CFA) of each item on the final HDMC Model.** F1 = Work Priority Strain, F2 = Physician's Moral Obligation to Patients and Colleagues, F3= Colleagues' Stigma towards PB, F4 = Personal Stigma towards PB, F5 = Physician's Discomfort with Patient Role; F6 = Physician's Central Identity Role, F7= Sacrificial Nature of Medical Practice (N = 1002). **Table I. First and second block of the hierarchical multiple regression of the association between the investigated dimensions of medical culture and burnout, while controlling for covariates.** Dependent variable is the PB score. The following covariates were controlled in the model in step 1: gender (female referent), relationship status (being in a relationship referent), parental status (having children referent), physician status (resident referent), specialty (specialist referent), years in practice, working hours estimated per week. In step 2, the HDMC and Existential Significance were added. N = 964. Missing values were excluded listwise. Std B = standardized beta. CI = Confidence Interval. By convention, we do not provide standardized betas for the constant of the regression. *Without* the factor Existential Significance of Being a Physician, the three second-order HDMC dimensions significantly contribute to the prediction of burnout, while controlling for the covariates ($R^2$ = 0.21). **Table J. Hierarchical multiple regression of the association between the investigated dimensions of medical culture and burnout, while controlling for covariates including social desirability.** Dependent variable is Burnout score. The following covariates were controlled in the model in step 1: gender (female referent), relationship status (being in a relationship referent), parental status (having children referent), physician status (resident referent), specialty (specialist referent), years in practice, working hours estimated per week. In step 2, the HDMC and Existential Significance were added. In step 3, we added social desirability. N = 934. Missing values were excluded listwise. In this regression, 16 cases were excluded as they were considered outliers (> |2| standard residuals) in the model. Std B = standardized beta. CI = Confidence Interval. By convention, we do not provide standardized betas for the

regression coefficient. Controlling for social desirability increased the proportion of variance in burnout explained by the model. This was unsurprising, as self-reported burnout may be impacted by the (un)desirability of suffering from burnout, especially so among physicians. Social desirability negatively predicted burnout (Std B = -0.24, $p$ <.001), and the three HDMC and Existential Significance remained significant predictors of burnout. Note that results including social desirability need to be interpreted cautiously as the internal consistency of this measure was low in our sample (α = 0.46). **Table K. Final HDMC Measure.** To obtain a score for each dimension separately, we averaged the items in each sub-dimension. To obtain an overall score for Harmful Dimensions of the Medical Culture (HDMC), we averaged all 21 items, where higher score indicates stronger internalization of the norms. This measure was developed in French language and has not yet been validated in an English-speaking sample. Items were back-translated from French to English.
(DOCX)

**S1 Fig.  Fig A. CFA model 1.** First-order model of *eight* distinct but correlated factors as extracted from the EFA, representing the concepts of Work Priority Strain; Existential Significance of Being a Physician; Physician's Moral Obligation to Patients and Colleagues; Colleagues' Stigma towards PB; Personal Stigma towards PB; Physician's Discomfort with Patient Role; Physician's Central Identity Role; and the Sacrificial Nature of Medical Practice and their indicators of 24 items. In this figure, the top left column represents the error (e) terms in the confirmatory factor analysis. The second column (rectangles) represents the items, the third column (arrows) represents the item loadings from the items to the first order latent factors, the fourth column (ovals) represents the first order latent factors found in the EFA, and the last column (arrows) represents the inter-factor correlations. **Fig B. CFA model 2.** First-order model of *seven* distinct but correlated factors extracted from the EFA, representing the concepts of Work Priority Strain; Physician's Moral Obligation to Patients and Colleagues; Colleagues' Stigma towards PB; Personal Stigma towards PB; Physician's Discomfort with Patient Role; Physician's Central Identity Role; and the Sacrificial Nature of Medical Practice and their indicators of 21 items. The top left column represents the error (e) terms in the confirmatory factor analysis. The second column (rectangles) starting from the left represents the items. The third column (arrows) represents the item loadings from the items to the first order latent factors. The fourth column (ovals) represents the first order latent factors found in the EFA. The last column (arrows) represents the inter-factor correlations. **Fig C. CFA model 3.** The second-order one-factor model in which the latent factors of Model 2 are supposed to measure harmful dimensions of the medical culture, loading onto a general factor for Harmful Dimensions of the Medical Culture (HDMC). The top left column represents the error (e) terms the confirmatory factor analysis. The second column (rectangles) starting from the left represents the items. The third column (arrows) represents the item correlations with the latent constructs for each item, the fourth column (ovals) represents the first-order latent factors found in the EFA, the fifth column (arrows) represents the loadings from the first order latent factors to the second-order latent HDMC factor, and the final column (ovals) represents the second order latent factor HDMC. **Fig D. CFA model 4.** Second-order hierarchical model where the *eight* initial factors of model 1 load onto three second-order dimensions (Physician's Professional Commitment; The Myth of the Invulnerable Physician; and Physician Stigma towards Burnout – *with* the latent factor of Existential Significance of Being a Physician). The top left column represents the error (e) terms in the confirmatory factor analysis. The second column (rectangles) starting from the left represents the 24 items. The third column (arrows) represents the item correlations with the latent constructs for each item, the fourth column (ovals) represents the first-order latent factors found in the EFA, the fifth column (arrows)

represents the loadings from the first-order latent factors to the
second-order latent dimensions, and the final column (ovals) represents the second-order
latent factors with inter-factor correlations (arrows). **Fig E. CFA model 5.** Second-order
hierarchical model where *seven* factors of model 1 load onto three second-order dimensions
(Physician's Professional Commitment; The Myth of the Invulnerable Physician; and Physi-
cian Stigma towards Burnout – *without* the latent factor of Existential Significance of Being a
Physician). The top left column represents the error (e) terms in the confirmatory factor anal-
ysis. The second column (rectangles) starting from the left represents the 21 items. The third
column (arrows) represents the item correlations with the latent constructs for each item, the
fourth column (ovals) represents the first-order latent factors found in the EFA, the fifth col-
umn (arrows) represents the loadings from the first-order latent factors to the second-order
latent dimensions, and the final column represents the second-order latent dimensions (ovals)
with inter-factor correlations (arrows).
(DOCX)

**S1 Text.**  Methods.
(DOCX)

## Acknowledgments

The authors are indebted to all participating medical organizations, editors of physicians'
magazines, Deans of medical faculties, medical directors and hospital councils for disseminat-
ing the survey. We gratefully acknowledge all physicians who participated to the online survey,
and also acknowledge Claire Ledouble, Alice Schittek, Aline Woine, Zoé Saliez, Melissa Salvra-
kos, Vénéthia Danthine, and Geraldine Petit (UCLouvain) for their insightful reviewing of the
items to measure the medical cultural dimensions. We thank Tom Taelemans from Crossword
for the translation of the items from French to English. We also thank the many physicians
who contacted us upon completion of the study to share their personal narratives of burnout
and distress and provided feedback on the survey. Finally, we authors thank Séverine Guisset
(SMCS - UCLouvain) for her help in the design of the Plan for Statistical Analyses.

We employed ChatGPT3.5 to refine the language of the manuscript using the command
"Below is an academic paper. Polish the writing to meet the academic style, improve spelling,
grammar, clarity, and overall readability." The authors express their indebtedness to Prof. Ron
Kupers (University of Copenhagen) for help in revising the paper, and Prof. Paul Cumming
(Bern University) for final corrections regarding language style.

## Author contributions

**Conceptualization:** Emilie Banse, Moïra Mikolajczak, Marie Bayot, Anne-Laure Lenoir,
   Philippe De Timary.

**Data curation:** Emilie Banse.

**Formal analysis:** Emilie Banse.

**Funding acquisition:** Emilie Banse.

**Investigation:** Emilie Banse.

**Methodology:** Emilie Banse.

**Project administration:** Emilie Banse.

**Resources:** Emilie Banse.

**Software:** Emilie Banse.

**Supervision:** Moïra Mikolajczak, Marie Bayot, Anne-Laure Lenoir, Philippe De Timary.

**Validation:** Moïra Mikolajczak, Marie Bayot, Anne-Laure Lenoir, Philippe De Timary.

**Visualization:** Emilie Banse.

**Writing – original draft:** Emilie Banse.

**Writing – review & editing:** Emilie Banse, Moïra Mikolajczak, Marie Bayot, Anne-Laure Lenoir, Philippe De Timary.

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
