## [Decision Letter · Decision Letter 0]

19 Nov 2024

PMEN-D-24-00354

Harmful dimensions of medical culture in relation to physician burnout: a cross-sectional study

PLOS Mental Health

Dear Dr. Banse,

Thank you for submitting your manuscript to PLOS Mental Health. After careful consideration, we feel that it has merit but does not fully meet PLOS Mental Health’s publication criteria as it currently stands. Therefore, we invite you to submit a revised version of the manuscript that addresses the points raised during the review process.

We look forward to receiving your revised manuscript.

Kind regards,

João Silvestre Silva-Junior, MD MSc PhD

Academic Editor

PLOS Mental Health

Journal Requirements:

Additional Editor Comments (if provided):

Reviewers' comments:

Reviewer's Responses to Questions

**Comments to the Author**

Reviewer #1: 

This study addresses the contribution of medical culture to physician burnout. Few data exist on this specific question, which is a relevant aspect in this field.

Title and Abstract: The study’s design is stated in the title, and the abstract provides a balanced summary.

Introduction: Previous literature was cited and treated fairly. The scientific background reports on recent studies. The specific objectives are stated.

Methods: Ethics and instruments are reported.

1. The study was carried out in a French-speaking country. Which validated translations of the instruments were used?

Population: The eligibility criteria and sources of participant selection are given. Convenience sampling is stated.

2. Period of recruitment is not given.

Statistical analyses are clearly described.

3. Outcomes, potential confounders, and effect modifiers are not clearly defined, nor are potential sources of bias.

4. Tables are informative. I guess it’s only in the French-speaking part of Switzerland that participants were included (if so specify).

Discussion: Key results are summarized, and limitations are discussed.

5. I didn’t see the Funding information statement.

6. The authors could consider using the STROBE checklist for cross-sectional studies and specify this in the manuscript.

7. The use of data from different countries (different cultures that may impact the local medical culture) could be discussed, as could the different types of hospitals and specialties. This could be discussed in the Discussion section.

Overall, the manuscript is well-organized and accessible to non-specialists.

Reviewer #2: 

General report:

The article addresses harmful dimensions in medical practice and their association with burnout among physicians, bringing a current and relevant topic into focus through an extensive review of the literature on the collective and individual factors shaping attitudes toward the practice of medicine. The proposal to validate a psychometric instrument for assessing these dimensions is innovative and represents the study's strongest contribution.

However, there is a disconnect between the main objective of the article — the validation of the instrument — and the way it is presented. Although the study is fundamentally psychometric, it is described as an observational study, which may confuse readers and divert attention from its primary contribution.

The cross-sectional data exploring the relationship between harmful dimensions and burnout are valuable for one stage of the instrument's validation, particularly for reliability analysis, but they are secondary to the study's main goal.

Specific Issues:

a) The introduction could be adjusted to better align with the journal's style. Beginning with a quote is not incorrect but may deviate from standard editorial practices. A more cohesive approach to introducing the problem is recommended.

b)Following the STROBE guidelines, it is essential to present the context and participant scenario (number of potential participants by country). If a census was conducted, the estimated participation percentage should be indicated.

While a sample size calculation is mentioned, insufficient detail is provided about the parameters used to determine the minimum sample size, which is critical for methodological transparency.

c) The fit indices used in the confirmatory (CFA) and exploratory factor analyses (EFA) are important and should be included in the methods section rather than being relegated to supplementary material.

d) The table 1 presents percentage data for explanatory variables for the total sample, but it also includes a specific subset (residents) and continuous variables, which makes it confusing to interpret. Suggested improvements include: Separating the general data from the subset (residents) with clear titles indicating the differentiation; Removing continuous variable values from the table, as the text already sufficiently presents medians and IQRs. Another advice is to make the table title more informative by detailing its content.

e) A particularly relevant issue that requires further explanation is the correlation of the second-order dimension C with the global HDMC score, which is considerably lower than the other dimensions. The authors should discuss possible reasons for this discrepancy, considering both theoretical and methodological factors.

---

## [Decision Letter · Decision Letter 1]

13 Mar 2025

Harmful dimensions of medical culture in relation to physician burnout: a cross-sectional study

PMEN-D-24-00354R1

Dear Ms Banse,

We are pleased to inform you that your manuscript 'Harmful dimensions of medical culture in relation to physician burnout: a cross-sectional study' has been provisionally accepted for publication in PLOS Mental Health.

Best regards,

João Silvestre Silva-Junior, MD MSc PhD

Academic Editor

PLOS Mental Health